# Promoting Fc-Fc interactions between anti-capsular antibodies provides strong immune protection against *Streptococcus pneumoniae*

Leire Aguinagalde Salazar[1], Maurits A den Boer[2,3], Suzanne M Castenmiller[1], Seline A Zwarthoff[1], Carla de Haas[1], Piet C Aerts[1], Frank J Beurskens[4], Janine Schuurman[4], Albert JR Heck[2,3], Kok van Kessel[1], Suzan HM Rooijakkers[1]*

[1]Medical Microbiology, University Medical Center Utrecht, Utrecht University, Utrecht, Netherlands; [2]Biomolecular Mass Spectrometry and Proteomics, Bijvoet Center for Biomolecular Research and Utrecht Institute for Pharmaceutical Sciences, Utrecht University, Utrecht, Netherlands; [3]Netherlands Proteomics Center, Utrecht, Netherlands; [4]Genmab, Utrecht, Netherlands

*For correspondence:
s.h.m.rooijakkers@umcutrecht.nl

**Abstract** *Streptococcus pneumoniae* is the leading cause of community-acquired pneumonia and an important cause of childhood mortality. Despite the introduction of successful vaccines, the global spread of both non-vaccine serotypes and antibiotic-resistant strains reinforces the development of alternative therapies against this pathogen. One possible route is the development of monoclonal antibodies (mAbs) that induce killing of bacteria via the immune system. Here, we investigate whether mAbs can be used to induce killing of pneumococcal serotypes for which the current vaccines show unsuccessful protection. Our study demonstrates that when human mAbs against pneumococcal capsule polysaccharides (CPS) have a poor capacity to induce complement activation, a critical process for immune protection against pneumococci, their activity can be strongly improved by hexamerization-enhancing mutations. Our data indicate that anti-capsular antibodies may have a low capacity to form higher-order oligomers (IgG hexamers) that are needed to recruit complement component C1. Indeed, specific point mutations in the IgG-Fc domain that strengthen hexamerization strongly enhance C1 recruitment and downstream complement activation on encapsulated pneumococci. Specifically, hexamerization-enhancing mutations E430G or E345K in CPS6-IgG strongly potentiate complement activation on *S. pneumoniae* strains that express capsular serotype 6 (CPS6), and the highly invasive serotype 19A strain. Furthermore, these mutations improve complement activation via mAbs recognizing CPS3 and CPS8 strains. Importantly, hexamer-enhancing mutations enable mAbs to induce strong opsonophagocytic killing by human neutrophils. Finally, passive immunization with CPS6-IgG1-E345K protected mice from developing severe pneumonia. Altogether, this work provides an important proof of concept for future optimization of antibody therapies against encapsulated bacteria.

## Editor's evaluation

This paper will be of interest to immunologists and infectious disease experts, as it reports the investigation of a novel treatment of invasive pneumococcal diseases using complement-activating monoclonal antibodies. Using a combination of in vitro and in vivo methods, the authors demonstrate convincingly that the introduction of specific mutations in human monoclonal antibodies that target

the surface of pneumococcus bacteria can result in enhanced complement activation after these antibodies bind to the bacterial surface.

## Introduction

The Gram-positive bacterium *Streptococcus pneumoniae* (pneumococcus) is the leading cause of community-acquired pneumonia and a major cause of bacteremia and meningitis in children and adults (*Loughran et al., 2019*; *Centers for Disease Control and Prevention website, 2019*; *O'Brien et al., 2009*). While pneumococcus commonly resides asymptomatically in the nasopharynx, it can cause a wide spectrum of infections in children, elderly, and immunocompromised patients (*van der Poll and Opal, 2009*; *Koedel et al., 2002*; *Birgitta and Tuomanen, 2013*; *Bryce et al., 2005*). Infections by pneumococcus range from non-invasive diseases, such as otitis media and sinusitis to life-threatening bacteremia and meningitis. To reduce its great impact on morbidity and mortality, vaccines have been successfully developed and introduced worldwide. Currently available vaccines target the polysaccharide capsule (CPS), which is considered the most important virulence factor of *S. pneumoniae*. Although there are more than 90 different capsular serotypes (*Bentley et al., 2006*), the current vaccines only include a limited number of serotypes including those most frequently found to be causing invasive pneumococcal disease (IPD). Besides the fact that the widespread vaccination has been highly effective in lowering IPD caused by vaccine serotypes (*Scott et al., 2009*; *Whitney et al., 2003*; *Shinefield and Black, 2000*), there still is a large burden of pneumococcal disease caused by non-vaccine serotypes. Furthermore, because some vaccine serotypes induce a weak immune response it remains difficult to control pneumococcal disease, particularly in risk groups (*Pirofski and Casadevall, 1998*). Finally, the emergence of strains with high level of antibiotic resistance (*McIntyre, 1997*; *Aguinagalde et al., 2015*; *Greenwood, 1999*; *Novak et al., 1999*) highlights a strong need to develop new therapeutic strategies against pneumococcal infections.

In recent years, antibody therapies have emerged as a successful treatment for several autoimmune diseases and cancers (*Scott et al., 2012*; *Edwards et al., 2004*). Therefore, there is now also great interest in the development of antibody-based therapies against bacterial infections. To eliminate bacteria, antibodies should bind to the bacterial surface and induce killing via the immune system. As evidenced by recurrent infections in patients with genetic complement deficiencies (*Ram et al., 2010*; *Carneiro-Sampaio and Coutinho, 2007*), human immune protection against pneumococci critically depends on the action of the human complement system (*Kaufmann and Dorhoi, 2016*; *Bardoel et al., 2014*). Complement is a large network of proteins in blood and other body fluids. These proteins circulate as inactive precursors but become rapidly activated upon contact with bacterial cells (*Wang et al., 2016*). An activated complement cascade triggers a variety of immune responses, such as the labeling of bacteria with C3-derived opsonins (C3b and iC3b) that potently induce phagocytosis and subsequent intracellular killing of bacteria by professional phagocytes (opsonophagocytic killing) (*Dunkelberger and Song, 2010*; *Stuart and Ezekowitz, 2005*).

Because complement is essential in immune protection against *S. pneumoniae* (*Kadioglu et al., 2008*; *Standish and Weiser, 2009*), the capacity of antibodies to induce complement activation could be exploited for effective antibacterial therapies. However, it is not known whether monoclonal antibodies effectively induce complement activation on *S. pneumoniae*. This is especially unclear for encapsulated *S. pneumoniae* strains because the capsule is believed to block immune activation, for instance, by shielding epitopes and blocking deposition of C3 opsonins (*Hyams et al., 2010*). Furthermore, recent studies showed that target-bound IgGs should organize into higher-order oligomers (IgG hexamers) to provide an optimal docking platform for complement component C1q (*Diebolder et al., 2014*; *de Jong et al., 2016*; *Figure 1a*, *Figure 1—figure supplement 1*). Because each antibody-binding headpiece of C1q has a low affinity for IgG, physiological binding occurs when the six headpieces simultaneously bind to IgG once hexamerized. Here, we study the potential of several anti-capsular monoclonal antibodies to induce complement activation and opsonophagocytic killing of encapsulated pneumococci. Our data suggest that these anti-capsular antibodies as wild-type IgG have a poor capacity to form IgG hexamers and trigger downstream opsonophagocytic killing via complement. Importantly, this limitation can be overcome by the introduction of a single amino acid mutation that enhances hexamerization of anti-capsular antibodies and strongly potentiates complement-mediated opsonophagocytic killing of pneumococci, both in vitro and in vivo.

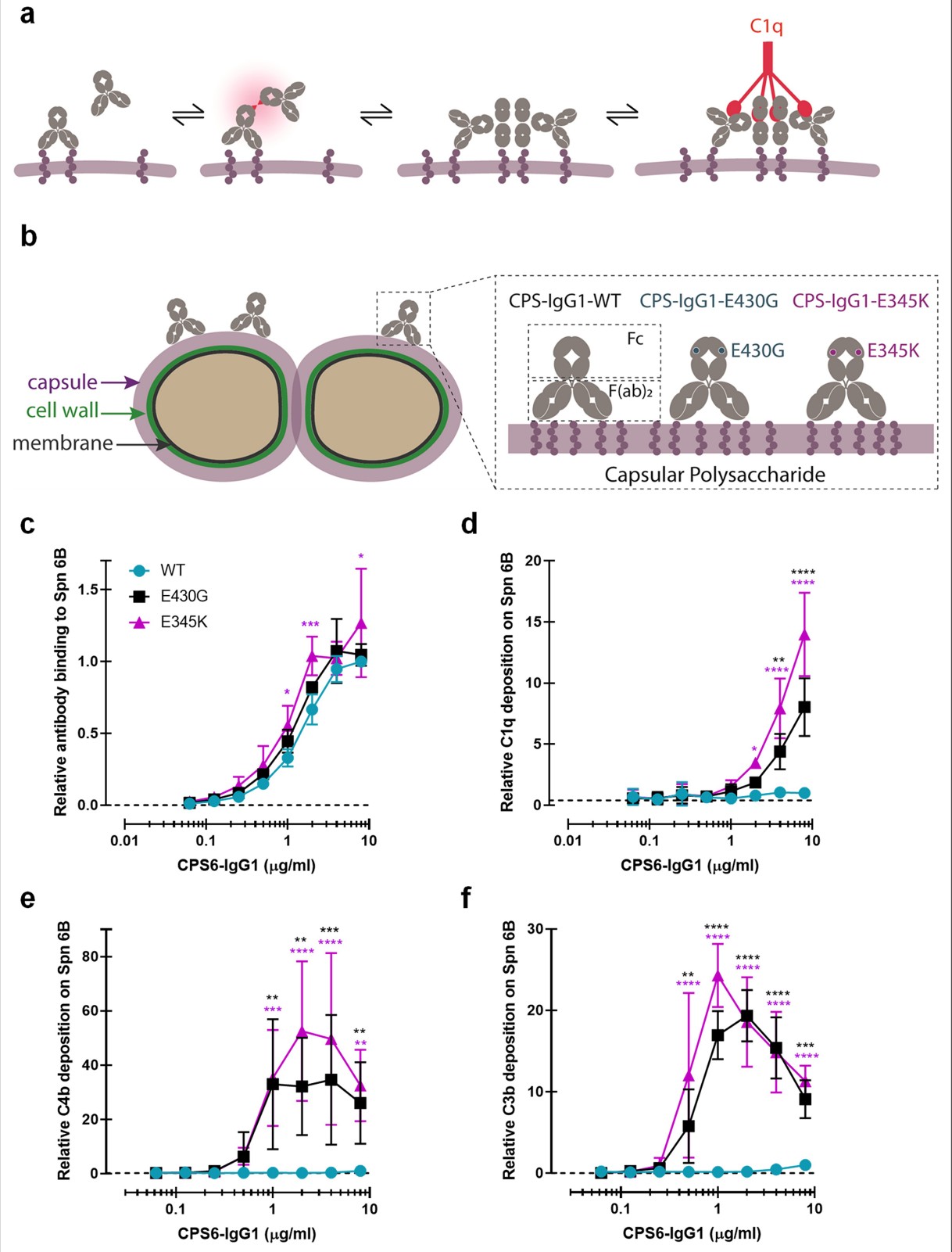

**Figure 1.** Promoting Fc-Fc interactions between CPS6-IgG1 enables complement activation on *S. pneumoniae* (Spn) 6B. (**a**) Schematic representation of antibody binding to antigen on a target surface. IgGs can cluster into hexamers via non-covalent interaction between their Fc domains and thus form an optimal docking platform for C1q. (**b**) Left: schematic illustration of *S. pneumoniae* showing the location of its dominant surface structure, the polysaccharide capsule (CPS), and antibodies recognition that confer type-specific protection. The capsule forms the outermost layer of encapsulated

*Figure 1 continued on next page*

*Figure 1 continued*

strains of *S. pneumoniae* and for most cases is covalently attached to the outer surface of the cell wall peptidoglycan. Right: binding of CPS-IgG1wild-type (WT) or containing the single-point hexamerization-enhancing mutations, E430G or E345K, to *S. pneumoniae* surface. (**c**) Binding of WT and hexamerization-enhancing mutated (E430G or E345K) CPS6-IgG1 to *S. pneumoniae* 6B (ST6B), detected with Alexa$^{647}$-conjugated F(ab')2-goat anti-human kappa antibody by flow cytometry. (**d–f**) Complement components C1q, C4b, and C3b deposition on *S. pneumoniae* 6B after incubation with 2.5% IgG/IgM-depleted serum supplemented with WT or hexamerization-enhancing mutated (E430G or E345K) CPS-IgG1. All detected with Alexa$^{647}$-conjugated F(ab')2-goat anti-mouse immunoglobulins antibody by flow cytometry. (**c–f**) Data are expressed relative to the 8 µg WT value and presented as means ± SD of three independent experiments. Dashed line represents background (no IgG) level. Two-way ANOVA was used to compare across dose–response curves at the various concentrations the differences between the WT and the E430G or E345K variants. When significant, it is displayed as *$p<0.05$; ***$p<0.001$; ****$p<0.0001$.

The online version of this article includes the following figure supplement(s) for figure 1:

**Figure supplement 1.** Overview of complement classical pathway activation.

**Figure supplement 2.** Representative flow cytometry histogram overlay showing equal binding of CPS6-IgG1 wild-type (WT), CPS6-IgG1-E430G and E345K mutants (4 µg/ml) to pneumococcal serotype 6B.

**Figure supplement 3.** Representative graph of *Figure 1c*.

**Figure supplement 4.** Representative graph of *Figure 1d*.

## Results

### Hexamer-enhancing mutations enable anti-capsular antibodies to activate complement on *S. pneumoniae* serotype 6B

To study whether human mAbs can induce complement activation on encapsulated *S. pneumoniae* strains, we first investigated complement activation on *S. pneumoniae* 6B, a common serotype infecting both adults and children (*Luján et al., 2010*; *Siber, 1994*). Although serotype 6B is covered by current vaccines, the 6B capsule type is found to be poorly immunogenic and an important risk factor in the mortality by IPD (*Luján et al., 2010*; *Siber, 1994*; *Sun et al., 1999*). We generated recombinant variants of a previously identified human IgG1 antibody that recognizes a carbohydrate structure present on serogroup 6 strains: α-D-Glcp(1→3)α-L-Rhap (*Park et al., 2009*). The variable, antigen-binding (Fab) domain of this polysaccharide serogroup 6-specific antibody (CPS6-IgG or 'Dob1'; *Sun et al., 1999*) was cloned into expression vectors containing the constant (Fc) domains of human IgG1. Furthermore, we introduced single amino acid mutations in the IgG1 Fc domain to enhance Fc-dependent hexamerization of target-bound antibodies (*Figure 1b*; *Wang et al., 2016*; *Diebolder et al., 2014*; *de Jong et al., 2016*). Specifically, mutations E430G (Glu$^{430}$ → Gly) or E345K (Glu$^{345}$ →Lys) were introduced because of their proven strong enhancement of complement-dependent lysis of tumor cells while retaining properties required for the development of biopharmaceuticals (*Diebolder et al., 2014*; *de Jong et al., 2016*).

After verifying that the introduction of hexamer-enhancing mutations did not affect the binding of CPS6-IgG1 antibodies to serotype 6B (*Figure 1c*, *Figure 1—figure supplements 2 and 3*), we studied their capacity to induce complement activation. To this end, 6B pneumococcus was incubated with mAbs in the presence of human serum as complement source. To exclude the involvement of pre-existing antibodies, we used human serum that was depleted from naturally occurring IgG and IgM (*Zwarthoff et al., 2021a*) (denoted IgG/IgM-depleted serum). Using surface-specific staining of C1q and flow cytometry, we first determined the capacity of mAbs to recruit C1q (*Figure 1—figure supplement 1*). While the wild-type (WT) CPS6-IgG1 showed little to no reactivity with C1q, we noted that the introduction of hexamer-enhancing mutations E430G or E345K strongly enhanced the ability to interact with C1q (*Figure 1d*). Importantly, we found that introduction of hexamer-enhancing mutations in CPS6-IgG1 allowed activation of the classical complement pathway. Recruitment of C1q to target-bound IgGs induces activation of C1q-attached C1r and C1s proteases that cleave C4 to covalently attach activated C4b molecules onto the bacterial surface (*Figure 1—figure supplement 1*; *Müller-Eberhard et al., 1967*). Furthermore, C1 activates C2 to produce a C3 convertase (C4b2b) that deposits large amounts of C3b, a key component of the complement cascade that labels bacteria for phagocytosis (*Brown et al., 1983*; *Joiner et al., 1984*). Indeed, by monitoring deposition of C4b and C3b molecules, we observed that Fc-engineered variants of CPS6-IgG1, but not the WT antibody, potently induced deposition of C4b (*Figure 1e*) and C3b molecules (*Figure 1f*,

*Figure 1—figure supplement 4*) onto serotype 6B. Altogether, these data show that hexamer-enhancing mutations can overcome poor complement activation by monoclonal antibodies against capsular serotype 6.

## Hexamer-enhancing mutations in CPS6-IgG enhance phagocytosis of *S. pneumoniae* serotype 6B

Next, we investigated whether the enhanced complement activation also impacted phagocytosis of serotype 6B by human neutrophils. Neutrophils are crucial to establish immune protection against pneumococcal infections (*Ullah et al., 2017*; *Lewis and Surewaard, 2018*). These cells are the first to be recruited from the blood to the site of infection where they engulf and internalize bacteria via phagocytosis to, subsequently, kill them by exposure to antimicrobial agents such as antimicrobial peptides, reactive oxygen species, and enzymes (*Bardoel et al., 2014*). The phagocytic uptake is greatly enhanced by the tagging of bacteria with IgG antibodies that can engage Fcγ receptors (FcγRs) (*Nimmerjahn and Ravetch, 2006*) and/or C3-derived opsonins that can mediate the uptake of bacteria via complement receptors (*Stuart and Ezekowitz, 2005*; *van Lookeren Campagne et al., 2007*). We studied phagocytosis of fluorescent serotype 6B by freshly isolated human neutrophils (*Bestebroer et al., 2007*) in the presence of CPS6-IgG1 mAbs and IgG/IgM-depleted human serum as complement source. First, we observed that the CPS6-IgG1-WT antibody poorly induced phagocytosis of serotype 6B (*Figure 2a*, *Figure 2—figure supplements 1 and 2*). In contrast, hexamer-enhanced variants of CPS6-IgG1 induced very potent phagocytosis (*Figure 2a*, *Figure 2—figure supplements 1 and 2*). For both E430G and E345K mutated variants, we found that 0.3 µg/ml of mAb induced maximum phagocytosis in the presence of 2.5% IgG/M-depleted serum (*Figure 2a*). When serum was heat-treated to inactivate complement (*Berends et al., 2015*), we found that phagocytosis by E430G and E345K antibodies was completely abolished (*Figure 2b*). The crucial role of complement in this process was also confirmed by microscopy (*Figure 2c*). To show that the flow cytometry assay really represents bacterial uptake of properly engulfed bacteria, confocal microscopy was performed. Opsonization with both E430G and E345K antibodies in the presence of active complement clearly resulted in bacterial internalization (*Figure 2d*, *Figure 2—figure supplement 3*). In contrast, opsonization by WT antibody resulted in mainly free extracellular localized bacteria (*Figure 2—figure supplement 3*). This indicates that phagocytic uptake via hexamerization-enhanced CPS6 antibodies is fully depended on the presence of complement.

To study whether phagocytosis via hexamer-enhanced antibodies indeed depends on Fc-dependent IgG oligomerization, we analyzed phagocytosis in the presence of Fc-III, a cyclic peptide that binds to IgG residues involved in the Fc-Fc interaction interface (*DeLano et al., 2000*). To study this end, we combined native mass spectrometry (*Leney and Heck, 2017*; *Tamara et al., 2022*) with an IgG triple mutant (IgG-RGY, combination of E345R (Glu$^{345}$ → Arg), E430G (Glu$^{430}$ → Gly), and S440Y (Ser$^{440}$ → Tyr) mutations) that has the capacity to form stable hexamers in solution (*Diebolder et al., 2014*). In agreement with prior work, native mass spectra of IgG1-RGY revealed the presence of monomeric and hexameric species, with intermediate states observed at lower abundance (*Figure 2e*; *Wang et al., 2016*; *Diebolder et al., 2014*). When IgG1-RGY was incubated with Fc-III, however, the relative abundance of IgG oligomers was diminished. No effect was observed for a scrambled version of the same peptide (Scr) that in contrast to Fc-III does not bind IgG1 molecules (*Figure 2e*, *Figure 2—figure supplement 4*). The fact that we observed binding of Fc-III to monomeric IgG1, but not to larger oligomeric species, suggests that Fc-III inhibits IgG-mediated complement activation by competitive binding to the Fc-Fc interaction interface of IgG monomers. In line with our hypothesis, we found that Fc-III potently blocked phagocytic uptake of serotype 6B via both E430G and E345K mutants, while Scr showed no effect (*Figure 2f*). Finally, considering that IgM antibodies already pre-exist as pentameric or hexameric oligomers that are kept together via covalent bonds, we compared the capacity of anti-CPS6-IgM to induce pneumococcal complement-mediated phagocytosis with our hexamerization-enhancing variants. Although IgM does induce complement-dependent phagocytosis of pneumococcal serotype 6B, it was less potent than CPS6-IgG1-E430G and E345K variants (*Figure 2g*, *Figure 2—figure supplement 5*). This suggests that complement activation via pre-assembled IgM oligomers was less effective than via IgG hexamers that are formed after target binding.

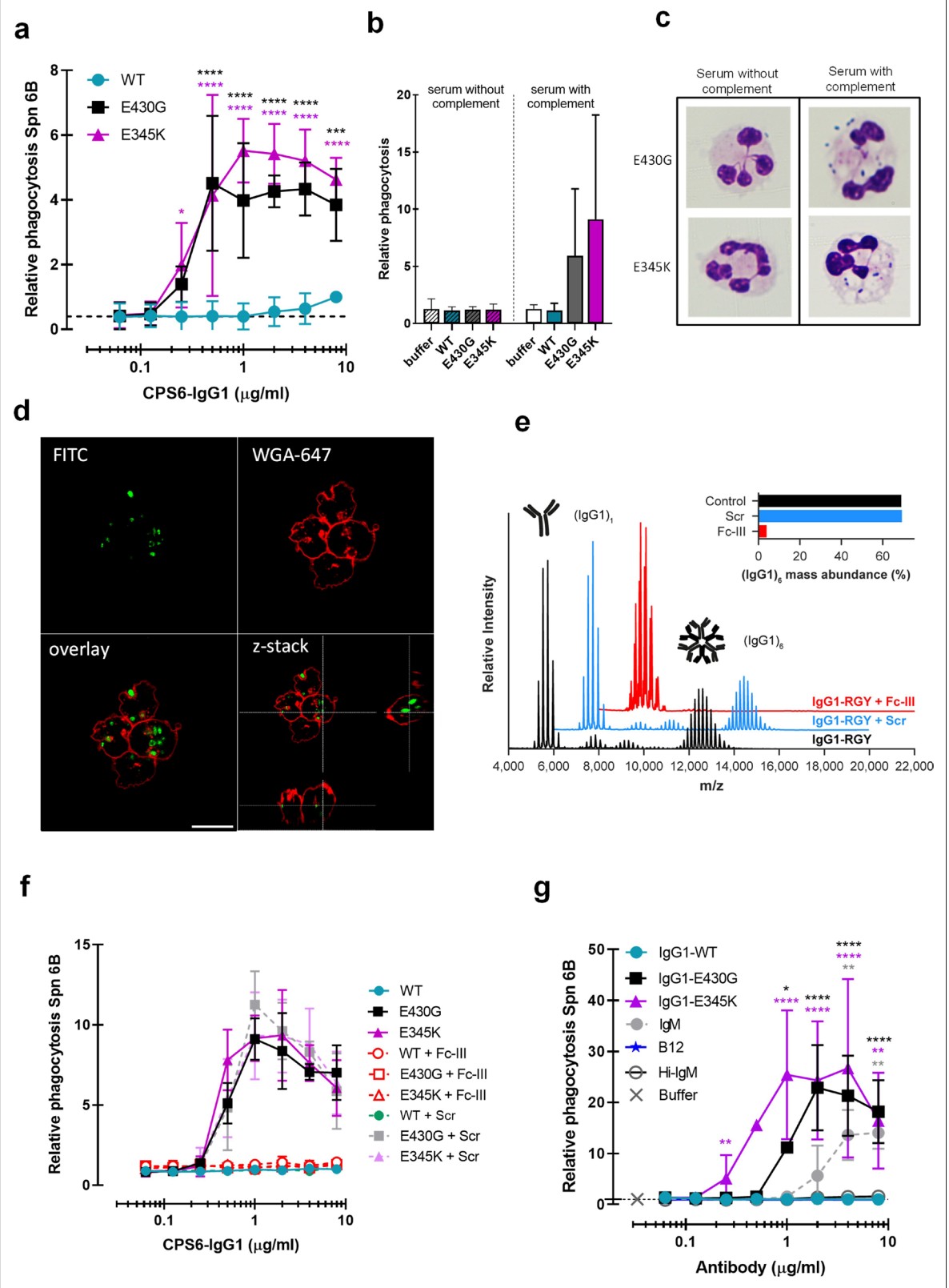

**Figure 2.** Hexamerization-enhanced variants of CPS6-IgG1 trigger complement-dependent phagocytosis of serotype 6B *S. pneumoniae*. (**a**) Phagocytosis in the presence of complement. Fluorescently labeled bacteria uptake by human neutrophils in the presence of 2.5% IgG/IgM-depleted serum supplemented with CPS6-IgG1-WT versus E430G and E345K variants. (**b**) Comparison of *S. pneumoniae* (Spn) serotype 6B phagocytosis by CPS6-IgG1-WT, E430G, or E345K antibody variants at 4 µg/ml in the presence of 2.5% IgG/IgM-depleted serum without (striped bars) or with active

*Figure 2 continued on next page*

*Figure 2 continued*

complement cascade (non-striped bars). (**c**) Microscopy image of pneumococcal phagocytosis by human neutrophils in the presence of 5% IgG/IgM-depleted serum with or without active complement, supplemented with 8 µg/ml CPS6-IgG1-E430G or E345K variant. Cytospin preparations were stained with Giemsa–May–Grünwald (Diff-Quik) and pictures taken using a ×100 objective to visualize cytoplasmic internalization. (**d**) Confocal microscopy images of *S. pneumoniae* 6B internalization by human neutrophils in the presence 2.5% IgG/IgM-depleted serum with active complement, supplemented with 4 µg/ml CPS6-IgG1-E345K variant. Bacteria were labeled with FITC (green) and neutrophils were visualized with WGA-Alexa 647 (red). Orthogonal view is representative for a total of three Z-stacks per condition. Scale bar: 10 µm. (**e**) Native mass spectra of IgG1-RGY in the absence (black) and presence of Fc-Fc inhibitor peptide Fc-III (red) or a scrambled version Scr (blue). Spectra are shifted for clarity and monomeric $(IgG1)_1$ and hexameric $(IgG1)_6$ mass peaks are indicated. Inset represents the percentage $(IgG1)_6$ for each sample. (**f**) Phagocytosis of fluorescently labeled *S. pneumoniae* 6B after incubation with 2.5% IgG/M-depleted serum supplemented with CPS6-IgG1-WT, CPS6-IgG1-E430G, or CPS6-IgG1-E345K in the presence or absence of 10 µg/ml Fc-Fc inhibitory peptide (Fc-III) and a scrambled version (Scr). (**g**) Phagocytosis of Spn 6B in the presence of anti-pneumococcal CPS6-IgM antibodies compared to CPS6-IgG1-WT, E430G, and E345K. Antibody b12 (anti-vitamin B12), which recognize HIV protein gp120, (B12) antibody was included as an unrelated negative control. (**a, b, f, g**) Bacterial uptake is displayed as the mean fluorescence value of neutrophils relative to CPS6-IgG1-WT at the highest concentration tested (8 µg/ml). Data represent mean ± SD of three independent experiment. (**a**) Dashed line represents background (buffer) level. Two-way ANOVA was used to compare across dose–response curves at the various concentrations the differences between the WT and the E430G or E345K variants. When significant, it is displayed as *p<0.05; ***p<0.001; ****p<0.0001.

The online version of this article includes the following figure supplement(s) for figure 2:

**Figure supplement 1.** Representative flow cytometry analysis of pneumococcal serotype 6B phagocytosis by neutrophils.

**Figure supplement 2.** Representative graph of *Figure 2a*.

**Figure supplement 3.** Confocal microscopy images of *S. pneumoniae* 6B internalized by human neutrophils in the presence of 2.5% IgG/IgM-depleted serum with active complement, supplemented with 4 µg/ml CPS6-IgG1-WT (**a**), E430G (**b**), or E345K variant (**b**).

**Figure supplement 4.** Deconvoluted native mass spectra showing that the masses of anti-CD52 IgG1 glycoforms (black) are shifted when incubated with Fc-III (red), but not with Scr (blue).

**Figure supplement 5.** Phagocytosis of *S. pneumoniae* (Spn) 6B in the presence of heat-inactivated 2.5% IgG/IgM-depleted serum supplemented with CPS6-IgG1-WT versus E430G, E345K variants or CPS6-IgM.

Taken together, while a WT monoclonal IgG antibody against CPS6 has a poor capacity to induce phagocytosis of *S. pneumoniae* serotype 6B, introduction of hexamerization-enhancing mutations strongly increases complement-mediated phagocytosis.

## Introduction of hexamer-enhancing mutations in IgG2 and IgG3 also results in increased pneumococcal recognition and clearance

Because the natural antibody response against bacterial capsule polysaccharides is dominated by IgG2 (*Siber et al., 1980*; *Ferrante et al., 1990*; *Schauer et al., 2003*), we also constructed human monoclonal CPS6-IgG2. While CPS6-IgG2-WT and Fc-Fc-enhancing variants showed equal binding to serotype 6B (*Figure 3—figure supplement 1a*), we again observed that hexamer-enhancing mutations E430G and E345K both improved C3b deposition in IgG2 (*Figure 3a*). Also for IgG3 antibodies, which are considered more effective in the induction of Fc-effector functions (*Vidarsson et al., 2014*), we found that the mutants had equal binding and hexamer-enhancing mutation E345K significantly increased C3b deposition, while a much less strong enhancement was observed for E430G (*Figure 3b*, *Figure 3—figure supplement 1b*). A direct comparison of E345K variants shows that CPS6-IgG1-E345K activated complement more potently than CPS6-IgG3-E345K and CPS6-IgG2-E345K (*Figure 3—figure supplement 1e*)**,** even though the binding is the same (*Figure 3—figure supplement 1c and d*).

Consistent with the results for C3b deposition, we observed that E345K, but not E430G, enhanced phagocytosis of CPS6-IgG3 (*Figure 3d*) while a very moderate effect was observed for IgG2 (*Figure 3c*). Again, phagocytosis of CPS6-IgG3 fully relied on the presence of active complement (*Figure 3—figure supplement 1f*).

## CPS6-IgG1-E430G and CPS6-IgG1-E345K induce potent complement activation and phagocytosis of serotypes 6A, 6C, and 19A

We wondered whether these results could be extended to other pneumococcal serotypes. Next to serotype 6B, the α-D-Glcp(1→3)α-L-Rhap antigen is also found in the CPS of serotypes 6A, 6C, and 19A but not in the 19F CPS (*Park et al., 2009*). Serotype 19A is of particular interest because this is a highly invasive serotype that, despite coverage in PCV-13 vaccine, remains one of the most frequently

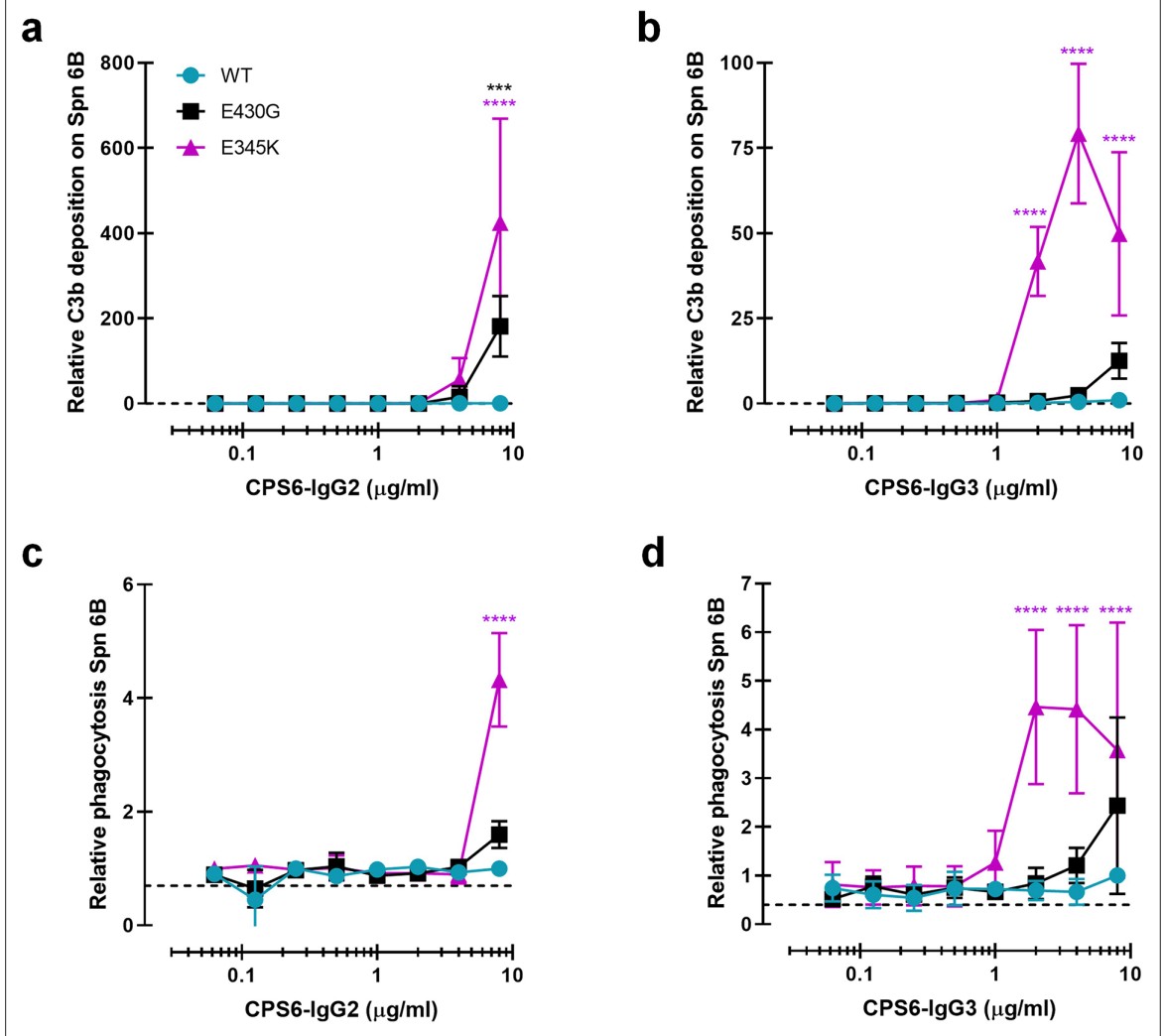

**Figure 3.** Introduction of E430G or E345K mutation in CPS6-IgG2 and CPS6-IgG3 improves complement activation and phagocytosis of *S. pneumonia* (Spn). (**a, b**) C3b deposition on serotype 6B surface after incubation of bacteria with CPS6-IgG2 (**a**) or CPS6-IgG3 (**b**) antibody variants in the presence of 2.5% IgG/IgM-depleted serum and detected with a monoclonal murine anti-human C3d antibody by flow cytometry. (**c, d**) Fluorescent serotype 6B bacterial phagocytosis by human neutrophils detected by flow cytometry after incubation in the presence of CPS6-IgG2 (**c**) or CPS6-IgG3 (**d**) hexamerization-enhanced variants, E430G and E345K, plus 2.5% IgG/IgM-depleted serum. All data are presented as mean fluorescence relative to the highest CPS-IgG-WT concentration tested (8 µg/ml). Dashed line represents background (no IgG) level. Data represent mean ± SD of at least two independent experiments. Two-way ANOVA was used to compare across dose–response curves at the various concentrations the differences between the WT and the E430G or E345K variants. When significant, it is displayed as *p<0.05; ***p<0.001; ****p<0.0001.

The online version of this article includes the following figure supplement(s) for figure 3:

**Figure supplement 1.** Comparison of IgG2 and IgG3 anti-CPS6 antibodies.

carried pneumococcal serotypes in children, and major cause of disease in European countries and the United States (*Kaplan et al., 2010*; *Isturiz et al., 2017*; *Bosch et al., 2016*; *Richter et al., 2013*). Furthermore, there is a significant increase in penicillin and multidrug resistance among 19A clinical isolates (*Richter et al., 2009*; *Beall et al., 2011*; *Ardanuy et al., 2009*; *Muñoz-Almagro et al., 2009*). After validating that (hexamer-enhancing variants of) CPS6-IgG1 indeed bind to serotype 19A but not to 19F (*Figure 4—figure supplement 1a and d*), we tested complement activation and phagocytosis of this strain. Similar to our results on serotype 6B, we observed that hexamer-enhancing mutations strongly increased complement activation on serotype 19A, as evidenced by increased detection of C1q (*Figure 4a*), C4b (*Figure 4b*), and C3b (*Figure 4c*), especially when E345K mutation was present. Consistently, CPS6-IgG1-E430G and CPS6-IgG1-E345K enhanced phagocytosis of *S. pneumoniae* serotype 19A phagocytosis in a dose-dependent manner (*Figure 4d*). Finally, we validated that

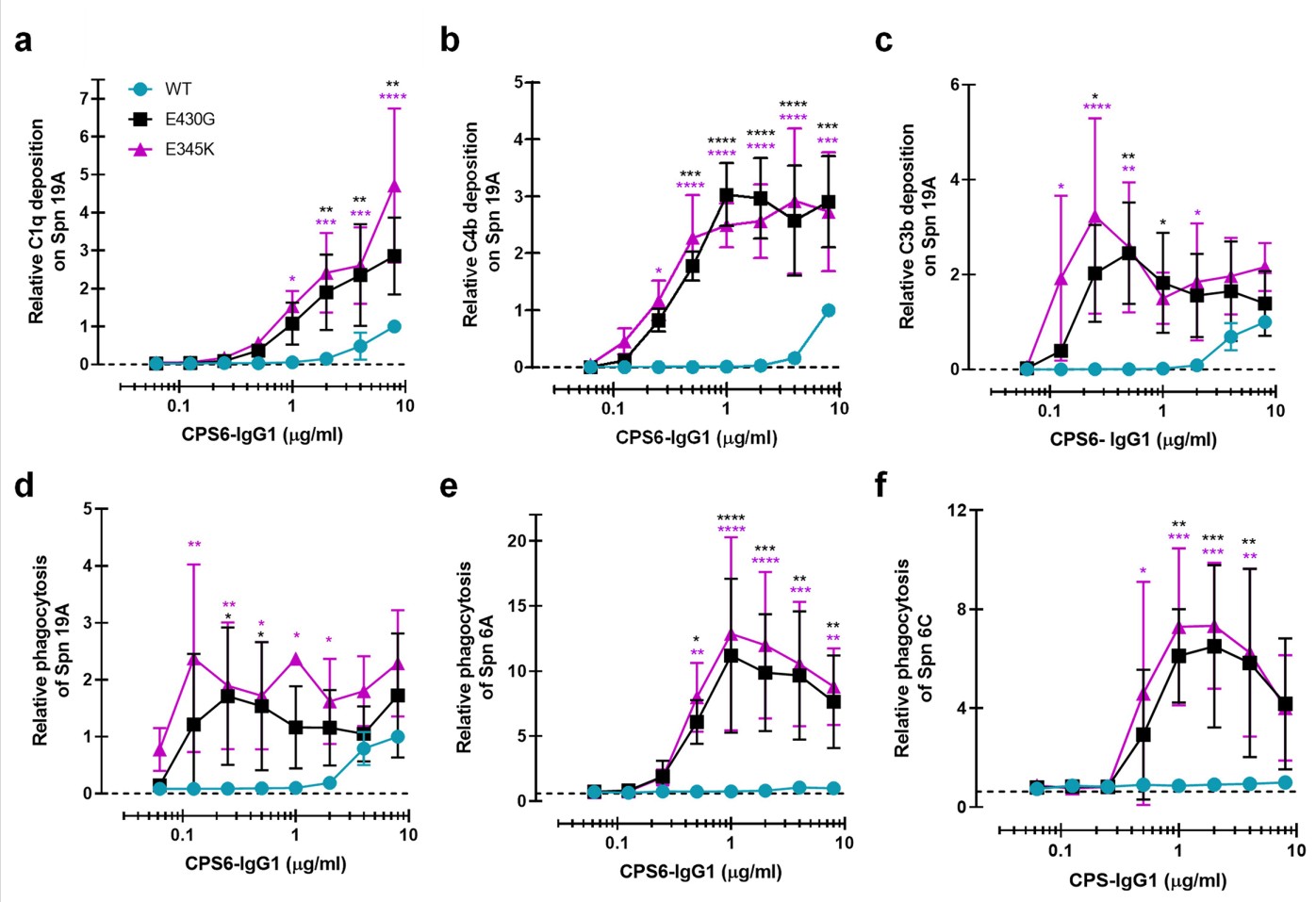

**Figure 4.** Enhanced Fc-Fc interactions strongly improves complement mediated phagocytosis of CPS6-IgG1-targeted *S. pneumoniae* serotypes. (**a–c**) Complement deposition on *S. pneumoniae* serotype 19A (Spn 19A) detected by flow cytometry after incubation with 2.5% IgG/IgM-depleted serum supplemented with CPS6-IgG1 (WT versus E430G and E345K variants). Detection of complement C1q (**a**), C4b (**b**), and C3b (**c**) deposition was done using a monoclonal anti-human C1q, C4d, or C3d antibody, respectively. (**d–f**) Phagocytosis of fluorescently labeled *S. pneumoniae* serotype 19A (**d**), serotype 6A (**e**), and serotype 6C (**f**) by human neutrophils in the presence of 2.5% IgG/IgM-depleted serum supplemented with CPS6-IgG1-WT versus E340G and E345K variants. Bacterial uptake was quantified by flow cytometry as the mean fluorescence of the neutrophils. All data represent relative mean ± SD of three independent experiments and displayed by the relative fluorescence index compared to CPS6-IgG1-WT at 8 µg/ml. Dashed line represents background (no IgG) level. Two-way ANOVA was used to compare across dose–response curves at the various concentrations the differences between the WT and the E430G or E345K variants. When significant, it is displayed as *p<0.05; ***p<0.001; ****p<0.0001.

The online version of this article includes the following figure supplement(s) for figure 4:

**Figure supplement 1.** Comparative binding of CPS6-IgG1 monoclonal antibody variants to *S.pneumoniae* serogroup 6 and 19.

hexamer-enhancing variants of CPS6-IgG1 also bind and improve complement-dependent phagocytosis of the poorly immunogenic serotype 6A (*Hostetter, 1986*; *Figure 4e*, *Figure 4—figure supplement 1b*) and the non-vaccine serotype 6C (*Figure 4f*, *Figure 4—figure supplement 1c*). Altogether, these results suggest that hexamer-enhancing variants of CPS6-IgG1 trigger complement-dependent phagocytosis of serogroup 6 pneumococci and the highly invasive serotype 19A.

## CPS6-IgG1-E430G and CPS6-IgG1-E345K induce opsonophagocytic killing of *S. pneumoniae* by human neutrophils in normal serum

Having established that hexamer-enhancing variants of CPS6-IgG1 strongly induce phagocytosis of serogroup 6 and serotype 19A strains, we studied whether these antibodies also trigger opsonophagocytic killing of bacteria by neutrophils. To mimic the natural situation more closely, we now performed experiments in normal human serum (NHS) that contains pre-existing antibodies (*Lebon*

et al., 2011; Borges et al., 2016) instead of using IgG/IgM-depleted serum as a complement source. This is important because 6B and 19A serotypes circulate in the healthy population and naturally occurring antibodies could thus play an additional role in phagocytosis by the mAbs. Previous experiments were repeated in the presence of normal serum as complement source. Indeed, E430G

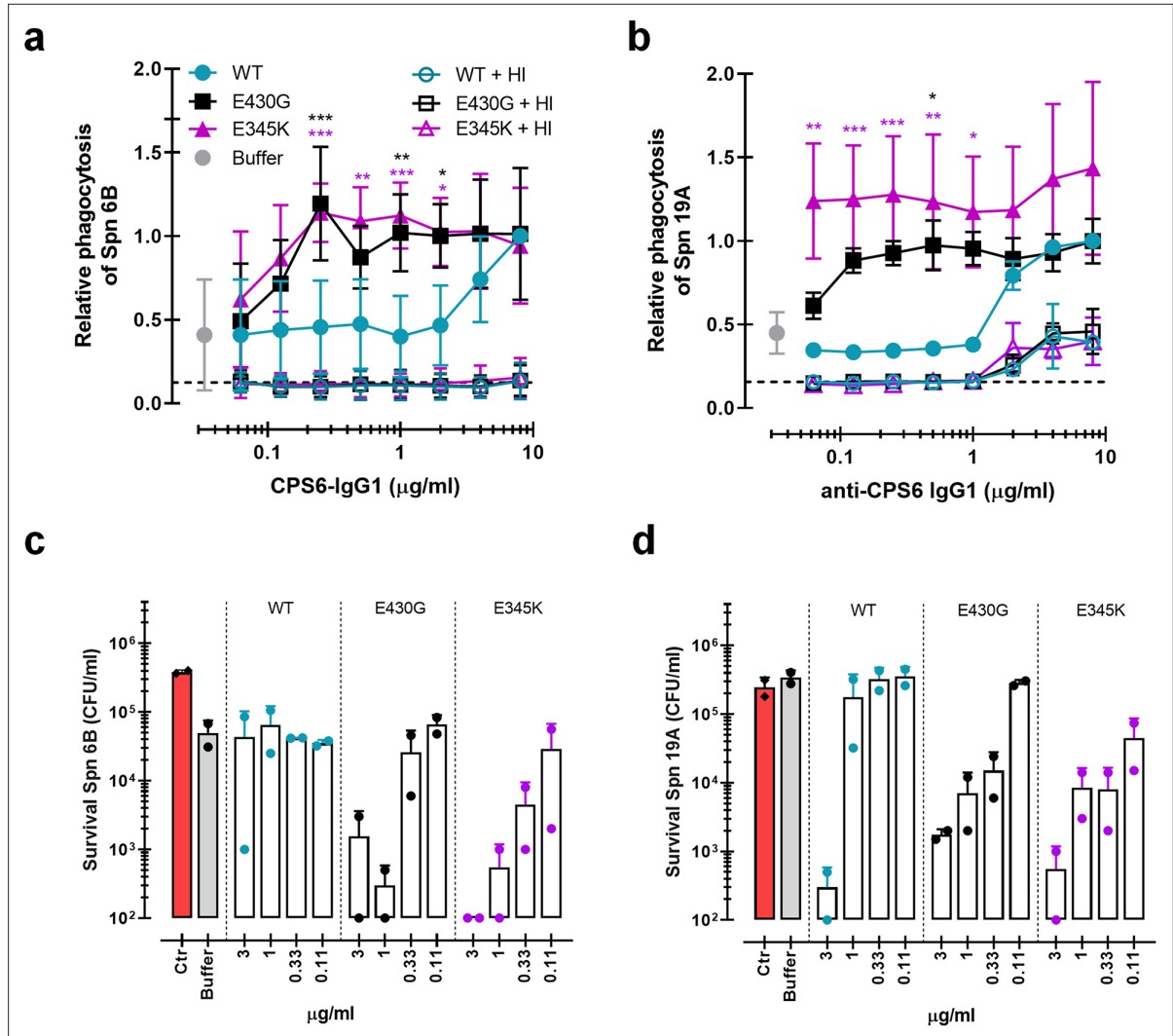

**Figure 5.** Monoclonal antibodies against *S. pneumoniae* capsule polysaccharide 6 can be modified for enhanced phagocytosis and opsonophagocytic killing by human neutrophils. (**a, b**) Phagocytosis by human neutrophils of fluorescently labeled *S. pneumoniae* (Spn) serotype 6B (**a**) or serotype 19A (**b**) in the presence of 5% normal human serum (NHS) as complement source supplemented with CPS6-IgG1-wild-type (WT) versus E430G and E345K hexamerization-enhanced variants. Same conditions for phagocytosis but in the presence of 5% NHS without complement activity (heat inactivated [HI]) is represented by empty colored boxes. Bacterial uptake was quantified by flow cytometry as the fluorescence of the neutrophils. Data represent relative fluorescence mean index ± SD of three independent experiments compared to the highest CPS-IgG1-WT concentration tested (8 µg/ml). Dashed line represents background (no IgG) level, and Buffer refers to the same condition with HPS (no IgG present). Two-way ANOVA was used to compare across dose–response curves at the various concentrations the differences between the WT and the E430G or E345K variants. When significant, it is displayed as *p<0.05; ***p<0.001; ****p<0.0001.(**c, d**) Opsonophagocytic killing of *S. pneumoniae* serotype 6B (**c**) or serotype 19A (**d**) in the presence of 5% NHS and CPS6-IgG1-WT versus CPS6-IgG1-E430G or CPS6-IgG1-E345K mutant. Bacterial survival was determined after 45 min incubation with human neutrophils by counting colony formation units (CFU) on blood agar plates. Red bars (Ctr) represent initial bacterial inoculum, whereas gray bars (Buffer) represent bacterial killing when antibodies were omitted. Data represent the mean ± SD of two independent experiments with duplicate counting (representing four experimental points).

The online version of this article includes the following figure supplement(s) for figure 5:

**Figure supplement 1.** CPS6-IgG1-E430G and E345K hexamerization-enhancing antibody variants potenciate complement deposition on *S. pneumoniae* (Spn) 6B in the presence of human normal sera.

**Figure supplement 2.** Opsonophagocytic killing of *S. pneumoniae* in human normal sera.

and E345K mutants also exhibited enhanced complement deposition (*Figure 5—figure supplement 1a and b*) and improved capacity to induce phagocytosis of serotype 6B (*Figure 5a*) and 19A (*Figure 5b*). In both cases, the presence of active complement was required as heat inactivation at 56°C completely abolished pneumococcal phagocytosis (*Figure 5a and b*). Next, to study whether the increased phagocytosis induced by antibody hexamerization results in effective bacterial killing by human neutrophils we performed opsonophagocytic killing assay (*Romero-Steiner et al., 1997*). To this end, bacteria were opsonized with CPS6-IgG1-WT or CPS6-IgG1-E430G/E345K antibody variants in the presence of normal human serum for 20 min. Then, human neutrophils were added for 45 min. Opsonophagocytic killing was measured by counting surviving colonies. Survival of serotypes 6B was strongly decreased in the presence of CPS6-IgG1-E430G or E345K in a dose-dependent manner, even achieving complete bacterial clearance at 3 µg/ml with E345K (*Figure 5c*, *Figure 5—figure supplement 2a*), confirming that the killing of *S. pneumoniae* by neutrophils after uptake is an efficient process. When the effectiveness of the CPS6-IgG1 hexamer-enhancing antibodies to improve killing of its cross-reactive serotype 19A was assessed few, if any, bacteria were recovered (*Figure 5d*, *Figure 5—figure supplement 2b*). These data show the functional positive consequences of antibody hexamerization on pneumococcal killing and their broader use among cross-reactive serotypes.

Overall, these results clearly show that mAb modification to induce hexamer formation on the bacterial surface potently increases complement-mediated *S. pneumoniae* phagocytosis and an effective intracellular killing.

## CPS6-IgG1-E345K engineered mAb for enhanced hexamerization protect mice against invasive pneumococcal infection

Following colonization of the nasopharynx, *S. pneumoniae* has the potential to invade the body and cause a broad spectrum of life-threatening diseases such as bacteremic pneumonia and meningitis. Transmission, colonization, and invasion depend on the remarkable ability of this bacteria to evade the host inflammatory and immune responses (*Weiser et al., 2018*). Hence, we investigated whether passive immunization with anti-capsular mAbs could protect mice against *S. pneumoniae* in a bacteremic pneumonia model. To this end, female BALB/c mice received an intraperitoneal injection of CPS6-IgG1-WT, CPS6-IgG1-E345K, or PBS (*Figure 6a*). Then, 3 hr later, pneumonia was induced by intranasal challenge with pneumococcal serotype 6A (selected due to its higher virulence in mice; *Saeland et al., 2000*). Survival and bacterial loads in the blood were monitored for 7 days (*Figure 6*, *Figure 6—figure supplement 1*). The presence of mAbs in the bloodstream was confirmed by ELISA (*Figure 6—figure supplement 1c*). In PBS-treated controls, all mice developed bacteremia within 24 hr as evidenced by the presence of high loads of pneumococci in the bloodstream (*Figure 6b*, *Figure 6—figure supplement 1a*). Passive administration with CPS6-IgG1-E345K effectively prevented bacterial dissemination to the bloodstream within the first 24 hr (*Figure 6b*, *Figure 6—figure supplement 1a*). While 100 µg CPS6-IgG1-E345K could protect 60% of infected mice (12/20) from developing bacteremia, the same concentration of CPS6-IgG1-WT only reduced bacteremia in 27% of mice (4/15 mice) (*Figure 6b*). A threefold higher concentration of CPS6-IgG1-WT was needed to achieve the same protection as CPS6-IgG1-E345K. Upon following survival, we observed that 95% (19/20) of PBS-treated mice succumbed due to the infection (*Figure 6b and c*). In contrast, 100 µg CPS6-IgG1-E345K could rescue 50% of mice (10/20). Again, we observed that the CPS6-IgG1-E345K mAb was more potent in protecting mice than CPS6-IgG1-WT. Additionally, mice weight regain was very much associated with the protective capacity of each mAb (*Figure 6—figure supplement 1b*). The fact that CPS6-IgG1-WT had a protective capacity to induce bacterial clearance in vivo when it showed a minimal effect in vitro can possibly be explained by the increased capacity of this mAb to induce phagocytosis in the presence of mice sera (*Figure 6—figure supplement 2*). These data indicated that other factors in mouse serum might play a role in protection by antibodies. However, the relevance of complement system for protection was demonstrated because pneumococcal phagocytosis was completely abolished in the presence of heat-inactivated sera.

When repeating the same experiments in male BALB/c mice, we observed that 100 µg CPS6-IgG1-E345K and 300 µg CPS6-IgG1-E345K could not confer significant protection against bacteremia (*Figure 6—figure supplement 3*). However, these mAbs were able to control bacterial spread from lungs to the bloodstream within 24 hr after infection in the same proportion as in female mice

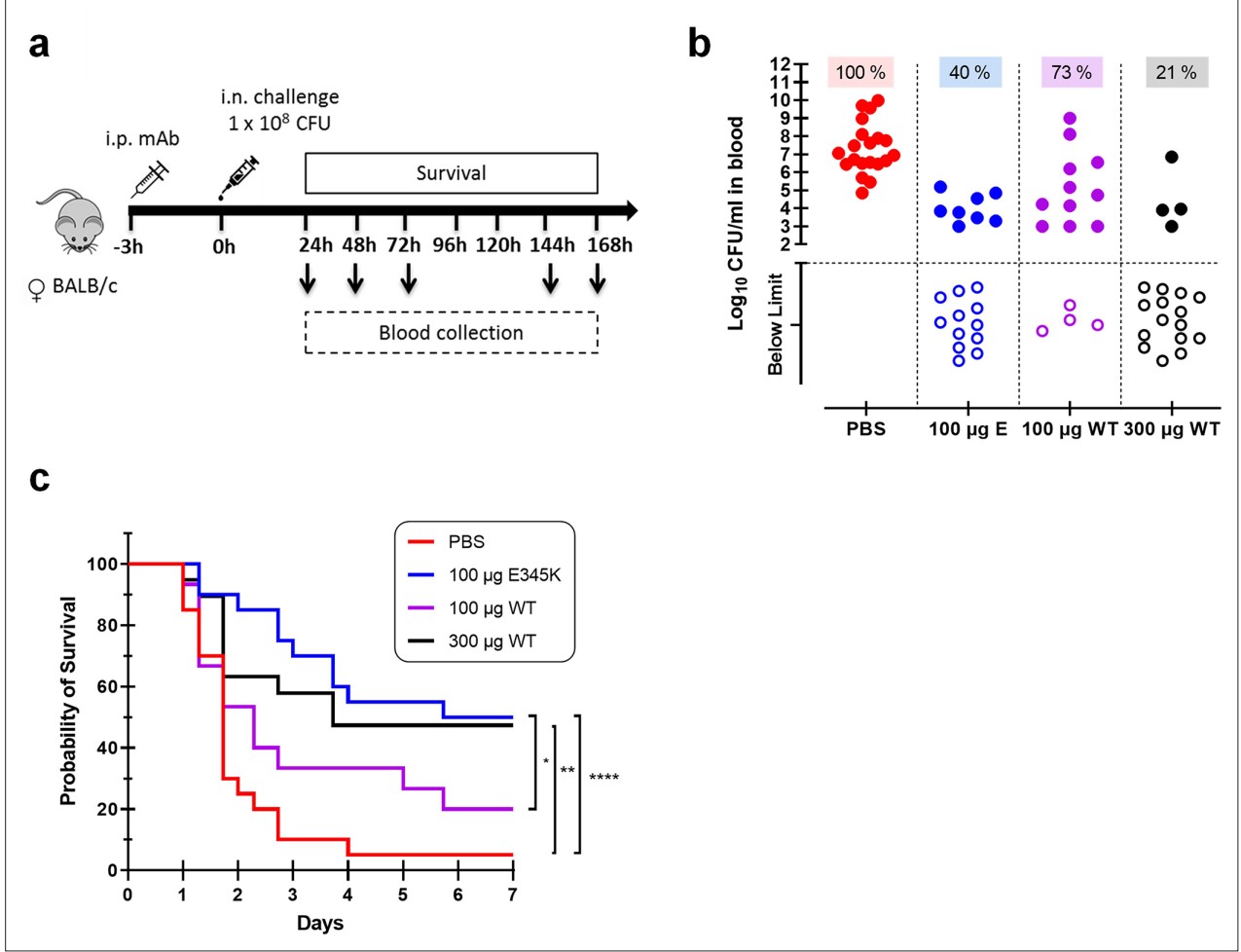

**Figure 6.** CPS6-IgG1-E345K mAb provides enhanced protection against invasive pneumococcal infection in mice. (**a**) Schematic representation of the infection model. Female BALB/c mice were passively immunized via intraperitoneal injection (i.p.) with PBS, 100 µg CPS6-IgG1-E345K, or 100 or 300 µg CPS6-IgG1-WT monoclonal antibody (mAb) 3 hr before infection. Mice were challenged intranasally (i.n.) with $1 \times 10^8$ CFU *S. pneumoniae* serotype 6A in 50 µl PBS. Every day after challenge, blood was taken from the tail vein, serially diluted, and plated on blood agar plates for bacterial colony counting (CFU). (**b**) Bacteremia in mice blood 24 hr after bacterial challenge representing mAb capacity to control bacterial spread from lungs to the systemic circulation (PBS 100%, 20/20; 100 µg E345K 40%, 8/20; 100 µg WT 73%, 11/15; 300 µg WT 21%, 4/15). Each symbol represents an individual mouse, closed symbols represent mice that developed bacteremia, and open symbols represent mice below the threshold of CFU detection marked by the dotted line. Mice survival was monitored in parallel for 7 days (**c**). The data are combined from three separate experiments with 5–8 mice for each treated group in each experiment resulting in 20 mice per group (only 15 mice for the 100 µg CPS6-IgG1-WT group). Statistical analysis was performed using log-rank (Mantel–Cox) test and is displayed when significant as *p≤0.1, **p≤0.01, or ****p≤0.0001.

The online version of this article includes the following figure supplement(s) for figure 6:

**Figure supplement 1.** Female BALB/c mice protection model.

**Figure supplement 2.** Human CPS6-IgG1 antibody variants induce complement-dependent phagocytosis of serotype 6B *S. pneumoniae* in the presence of mouse sera.

**Figure supplement 3.** Protective efficacy in male mice of engineered CPS6-IgG1-E345K.

**Figure supplement 4.** Local protection capacity of engineered CPS6-IgG1-E345K in mice.

---

(*Figure 6b*). Although this needs further investigation, these data potentially indicate gender differences in immune protection to pneumococcal infections.

Finally, we addressed the ability of the administered mAbs to clear the infection locally in the lungs. To do so, a group of mice (n = 12; six females and six males) were immunized and infected following the same procedure but animals were euthanized 24 hr post-infection. Then, bacterial load in mice BALF, lungs, and blood was determined by CFU counting. The fact that we did not find bacterial

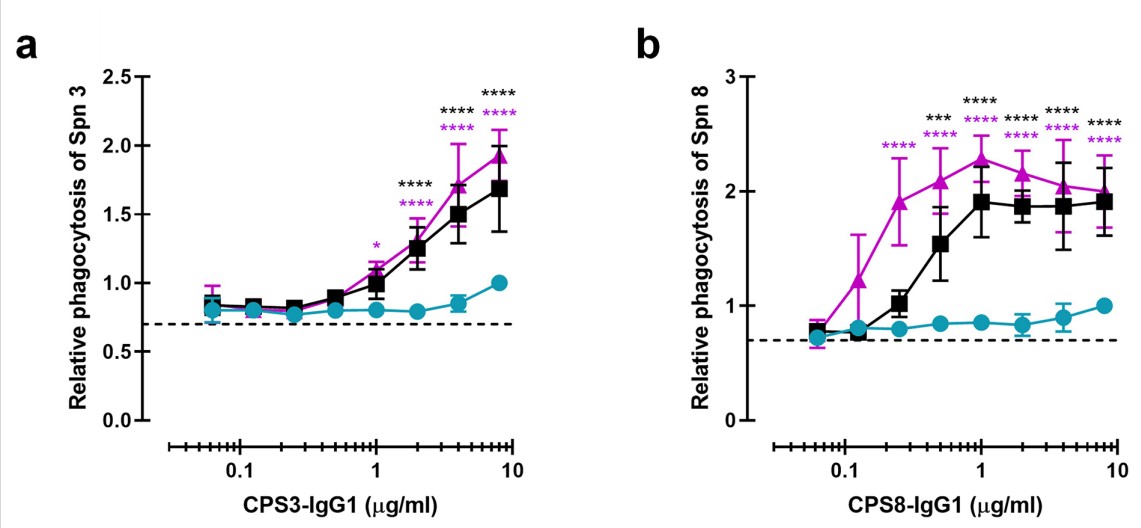

**Figure 7.** Monoclonal antibodies against CPS can be modified for enhanced complement-mediated pneumococcal phagocytosis. (a, b) Phagocytosis by human neutrophils of fluorescence-labeled *S. pneumoniae* (Spn) serotype 3 (**a**) and serotype 8 (**b**) after incubation with 5% human normal sera supplemented with CPS-IgG1-WT, CPS-IgG1-E430G, or CPS-IgG1-E345K variant. All data represent relative fluorescence mean index ± SD of three independent experiments compared to the highest CPS-IgG1-WT concentration tested (8 µg/ml). Dashed line represents background (no IgG) level. Two-way ANOVA was used to compare across dose–response curves at the various concentrations the differences between the WT and the E430G or E345K variants. When significant, it is displayed as $*p<0.05$; $***p<0.001$; $****p<0.0001$.

The online version of this article includes the following figure supplement(s) for figure 7:

**Figure supplement 1.** Comparative binding of CPS3 and CPS8-IgG1 antibody variants to *S. pneumoniae* strains.

load differences between CPS6-IgG1-WT and CPS6-IgG1-E345K (100 µg) in mice lungs and BALF (*Figure 6—figure supplement 4*) suggests the effect of mAbs is not preventing bacteria from stabilizing an infection in the lungs, but mainly preventing bacterial dissemination to the bloodstream.

Altogether, these data suggest that passive immunization with complement-enhancing monoclonal antibodies can prevent mice from developing bacteremia by encapsulated pneumococci.

### Promoting Fc-Fc interactions enhances complement-dependent phagocytosis via mAbs recognizing CPS3 and CPS8

Being aware of the fact that CPS6 mAbs can only target a small fraction of all circulating pneumococci, we also studied phagocytosis of mAbs against other capsular serotypes. We constructed IgG1 antibodies against capsule serotype 3 or 8 using variable region sequences of these mAbs available from literature (*Tian et al., 2009*; *Yano and Pirofski, 2011*). Serotype 3 causes disease in both adults and children and has been associated with an increased risk of death compared to other pneumococcal serotypes (*Tian et al., 2009*; *Martens et al., 2004*). Unlike serotype 3, serotype 8 is not included in PCV vaccines, and therefore its prevalence remains increasing (*Duvvuri et al., 2016*; *Rodríguez et al., 2011*) worldwide. After verifying mAbs variants equal binding (*Figure 7—figure supplement 1*), we studied complement-mediated serotype 3 and 8 phagocytosis in the presence of 5% NHS and the CPS3 or CPS8 mAb variants (*Figure 7a and b*). Once more, we found that hexamerization-enhancing mutations highly enhanced phagocytosis of both serotypes 3 and 8 in a dose-dependent manner.

Overall, these results confirmed that antibodies directed against CPS could be used as therapy target as they can be modified to enhance hexamer formations, having strong positive influence on complement deposition and phagocytosis of highly virulent *S. pneumoniae* strains.

## Discussion

Here, we studied whether monoclonal antibodies directed against the capsule of *S. pneumoniae* have the potency to induce bacterial killing via the human immune system. Although our data indicate that WT anti-capsular IgGs used in this study had a poor capacity to induce elimination of bacteria, the

introduction of hexamer-enhancing mutations enabled such mAbs to induce potent immune clearance of *S. pneumoniae* infection, both in vitro and in vivo.

Immune-mediated killing of Gram-positive bacteria strongly depends on the capacity of human neutrophils to engulf bacteria via phagocytosis and kill them intracellularly (*Standish and Weiser, 2009*; *Gingerich et al., 2023*). Antibodies can enhance this process by binding to the bacterial cell surface and stimulating FcγR uptake and/or activate the complement cascade to deposit C3-derived opsonins. Although the pneumococcal polysaccharide capsule is known to evade complement activation (by shielding epitopes that are hidden underneath the capsule; *Hyams et al., 2010*), our results support the idea that antibodies directed against the capsule can overcome capsular immune evasion and deposit complement opsonins on the encapsulated bacterium to stimulate phagocytosis. For the anti-capsular antibodies included in this study, we showed that monoclonal IgGs had a poor capacity to activate complement when expressed with wild-type Fc domains. At present, it is unclear why these WT anti-capsular antibodies showed a poor capacity to induce complement activation on *S. pneumoniae*. We are confident that this is not related to antibody production issues. Previously we have generated and functionally characterized monoclonal IgG1/3 antibodies against *S. aureus* wall teichoic acid (WTA) (*Zwarthoff et al., 2021b*). The cloning, expression, and purification procedures of these anti-WTA antibodies were identical to the antibodies used in this article (they even have the same IgG-Fc backbone). In contrast to this study, the wild-type IgG1/IgG3 antibodies targeting WTA were capable of inducing complement activation and downstream phagocytosis in the absence of Fc-enhancing mutations. Therefore, we believe that the anti-capsular antibodies used in this study have a poor ability to establish Fc-Fc contacts needed for hexamer formation. Potentially, the capsule epitope is not an optimal antigen for IgG hexamer formation. For instance, molecular characteristics like antigen density, size, and fluidity could affect hexamer formation (*Diebolder et al., 2014*). Also, the affinity of one Fab arm for the epitope could play a role since antibodies should bind with one Fab arm to enable hexamer formation (*Diebolder et al., 2014*). Whether the observed low capacity of wild-type CPS-specific mAb to induce IgG clustering can be extended to other anti-CPS or anti-pneumococcal antibodies would need further investigation. Potentially, our data may help to understand why certain capsular serotypes are poorly immunogenic. CPS antigen is poorly immunogenic in many individuals who are at the highest risk for the development of IPD and the use of protein conjugated vaccine has not been found in these patients to be more immunogenic than the purified polysaccharide-based vaccine (*Chang et al., 2002*). Since complement activation and deposition of C3-derived opsonins are important in driving adaptive immune responses to bacteria, for instance, by enhancing B cell activation and antigen presentation (via complement receptors on B cells, APCs, and follicular dendritic cells) (*Bennett et al., 2017*), a poor capacity to induce antibody clustering might also have a negative effect on vaccine efficacy.

Our findings also indicate that phagocytic uptake of *S. pneumoniae* via CPS-specific antibodies strongly depends on the presence of complement-induced opsonization since we observed little to no uptake of antibody-labeled bacteria in the absence of complement. This reinforces the vital role of complement in pneumococcal clearance, which is also demonstrated by the fact that patients with complement deficiencies are at high risk of *S. pneumoniae* infections (*Carneiro-Sampaio and Coutinho, 2007*; *Jönsson et al., 2006*). Our finding that hexamer-enhancing mutations can boost complement-dependent killing of bacteria by itself is not new since these same mutations have been demonstrated to potentiate complement-dependent killing of mAbs targeting *Neisseria gonorrhoeae* (*Gulati et al., 2019*). However, complement-dependent killing of *Neisseria* is mediated by the formation of membrane attack complex pores, which cannot act on Gram-positive bacteria (*Berends et al., 2014*). Our study provides an important proof of concept that hexamerization-enhancing mutations can potentiate opsonophagocytic killing of Gram-positive bacteria. This assumption is also supported by our in vivo bacteremic pneumonia mice model of infection. The presence of CPS-IgG1 mAbs potently prevented/delayed pneumococcal spread from lungs to the systemic circulation, resulting in effective clearing and, therefore, conferring protection against *S. pneumoniae* infection in female mice. The obtained protection was strongly increased by the use of engineered mAb CPS6-IgG1-E345K for enhanced hexamerization.

Although this was not the goal of our study, the in vivo experiments also revealed that the monoclonal antibodies provided less protection against pneumococcal bacteremia in male than female mice. Clinical manifestation of the infection and mortality rate was comparable between male and

female for the administered infectious dose. Dutch legislation encourages the verification of both genders in animal experiments, both for animal welfare and impact on translation to human diseases. Whether the immune responses to pneumococci and response to antibody therapies are truly different between males and females is beyond the scope of this article but an important consideration for future therapy developments.

Overall, the presented work provides a proof of concept not only for the capacity of hexamer-enhancing mutations to improve anti-capsule mAb-meditated immune system activation, but also for the potential use of monoclonal antibodies against encapsulated bacteria like *S. pneumoniae* when the existing therapies fail. In this study, we demonstrated that the activity of mAbs directed against serogroup 6, one of the most prevalent serogroups worldwide (*Shi et al., 2018*; *van der Linden et al., 2013*), could be effectively improved to prevent systemic pneumococcal infection. Furthermore, the fact that CPS6 mAbs likely cross-react with the highly invasive 19A strain, for which of invasive infection rates increased following PCV7 use worldwide (*Aguiar et al., 2010*), suggests the potential broader use of monoclonal antibodies. Similarly, hexamer-enhancing mutations could be used to improve the potency of recently discovered mAbs against other serotypes like CPS3 (*Babb et al., 2021*). Because complement is essential in immune protection against *S. pneumoniae* (*Kadioglu et al., 2008*; *Standish and Weiser, 2009*), the capacity of antibodies to induce complement activation could be exploited for effective antibacterial therapies while simultaneously avoiding the complications of antibiotic resistance. However, the large variety in pneumococcal CPS serotypes hampers the use of serotype-specific mAbs to treat pneumococcal disease. Therefore, the most clinically useful scenario for the use of anti-pneumococcal CPS mAbs is to treat after symptoms onset. A cocktail composed of multiple anti-capsule mAbs would be very expensive unless, for instance, rapid test to identify the causative serotype can be developed to enable monovalent therapy. Monoclonal antibodies that react with highly conserved surface antigens that elicit a potent immune response like histidine triad protein D (PhtD) (*Gingerich et al., 2023*) can be a promising tool for broad treatment against numerous pneumococcal serotypes. From the presented results, we anticipate that this work will stimulate new routes to optimize antibody therapies against *S. pneumoniae* and other encapsulated bacteria.

## Materials and methods
### Bacterial strains and fluorescent labeling
*S. pneumoniae* clinical isolates used in this study included strains from serotype 3, 6A, 6B, 6C, 8, 19A, and 19F (kindly provided by Dr. J. Yuste, Centro Nacional de Microbiología, CNM-ISCIII, Madrid). Pneumococcal isolates were cultured on blood agar plates at 37°C in 5% $CO_2$ or in Todd–Hewitt broth medium supplemented with 0.5% yeast (THY) extract to an optical density at 550 nm ($OD_{550}$) of 0.6 and stored at –80°C in 10% glycerol as single-use aliquots for further experiments. For generation of fluorescently labeled bacteria, strains were grown in THY, washed with PBS, and incubated with fluorescein isothiocyanate (FITC; Sigma) (0.5 mg/ml) for 60 min on ice in dark. Bacteria were washed four times with PBS, resuspended in RPMI-0.05% human serum albumin (HSA), and stored as aliquots in 10% glycerol at –80°C.

### Isolation of human serum
Human blood from healthy volunteers was collected in plastic vacuette tubes (Greiner) with informed consent in accordance with the Declaration of Helsinki. Approval from the Medical Ethics Committee of the University Medical Center Utrecht was obtained (METC protocol 07-125/C approved on March 1, 2010). Clotting was allowed for 15 min and blood was centrifuged at 3166 × *g* for 10 min at 4°C to collect serum. Sera of 20 donors were pooled and stored as single-use aliquots at –80°C as a source of complement and serum components. As an alternative complement source, the same NHS was depleted for IgG and IgM using a HiTrap Protein-G and Poros anti-IgM column in tandem on an Akta FPLC system (GE-Healthcare) as previously described (*Zwarthoff et al., 2021a*). The minimum loss of complement activity has been proven for this procedure (*Zwarthoff et al., 2021a*).

## Human monoclonal antibody production

Human monoclonal antibodies were produced recombinantly in human Expi293F cells (Life Technologies) as described before (*Cruz et al., 2021*), with minor modifications. Briefly, gBlocks (Integrated DNA Technologies [IDT]), containing codon-optimized variable heavy and light chain (VH and VL) sequences with an upstream KOZAK and HAVT20 signal peptide, were cloned into homemade pcDNA34 vectors, upstream the IgG/IgM heavy and kappa light chain constant regions, respectively, using Gibson assembly (New England Biolabs). The VH and VL sequences of the antibodies were derived from previously reported antibodies against CPS6 (*Saeland et al., 2003*), CPS3 (*Tian et al., 2009*), and CPS8 (*Yano and Pirofski, 2011*), with some modifications (*Supplementary file 1*). For IgM, a plasmid coding for J-chain expression was a kind gift from Theo Rispens. After verification of the correct sequence, the plasmids were used to transfect EXPI293F cells (Thermo Fisher Scientific). EXPI293F cells were grown in EXPI293 medium in culture filter cap Erlenmeyer bottles (Corning) on a rotation platform (125 rotations/min) at 37°C, 8% $CO_2$. One day before transfection, cells were diluted to $2 \times 10^6$ cells/ml. Transfection of EXPI293F cells was performed using PEI (Polyethylenimine HCl MAX; Polysciences). Therefore, 1 µg DNA/ml cells (ratio of heavy and light chain plasmids is 2:3) was added to OPTIMEM (1:10 of total volume; Gibco) and gently mixed. For expressions of IgM containing the J-chain, the J-chain plasmid was used as 20% of total plasmid. After adding PEI (1 µg/ml; ratio PEI to DNA is 5:1), the mixture was incubated at room temperature (RT) for 20 min and then added dropwise to the cells while manually rotating the Erlenmeyer culture bottle. After 4–6 days of transfection, IgG1 and IgG2 antibodies were isolated from cell supernatants using a HiTrap Protein A High Performance column (GE Healthcare), whereas IgG3 antibodies were isolated with a HiTap Protein G High Performance column (GE Healthcare). Antibodies were dialyzed in PBS, overnight at 4°C, and filter-sterilized through 0.22 µm Spin-X filters. IgG antibodies were analyzed by size-exclusion chromatography (GE Healthcare) and monomeric fractions were isolated in case of aggregation levels >5%. The concentration of the antibodies was determined by measurement of the absorbance at 280 nm and antibodies were stored at –20°C until use. For IgM, after 5 days of expression, the cell supernatant was collected by centrifugation and filtration (0.45 µM) and dialyzed against PBS. After dialysis, extra NaCl was added to the IgM preparation to a final of 500 mM before application to a POROS CaptureSelect IgM Affinity matrix (Thermo Scientific) column. IgM was eluted with 0.1 M glycine-HCl pH 3.0. on the ÄKTA Pure (GE Healthcare). 0.5 M NaCl was added to the pooled fraction, which was neutralized with 1 M Tris pH 7.5. IgM was dialyzed against PBS, containing 0.5 M NaCl. Finally, pentameric/hexameric IgM was isolated on a Superose 6 Increase 10/300 GL in PBS/500 mM NaCl and stored at –4°C or –80°C for long storage.

As negative control, we produced one antibody recognizing HIV protein gp120 (B12-IgG) (*Barbas et al., 1993*; *Saphire et al., 2001*).

Monoclonal IgG1 antibodies against CD52 (alemtuzumab; *Crowe et al., 1992*), recombinantly expressed as WT and hexamer-forming RGY mutant (*Diebolder et al., 2014*), were obtained from Genmab (Utrecht, the Netherlands).

## Antibody binding and deposition of complement components on bacterial surface

Antibody binding and complement C1q, C4b, and C3b deposition was assessed on FITC-labeled strains in RPMI-HSA using flow cytometry as previously described (*Yuste et al., 2008*). For anti-CPS binding, bacteria were incubated with antibody for 20 min at 4°C, washed, and incubated for another 30 min at 4°C with APC-labeled donkey-anti-human-IgG (Jackson ImmunoResearch Europe Ltd). For complement deposition assays, 2.5% IgG/IgM-depleted serum or 5% NHS was used. Bacteria were incubated with a concentration range of twofold serial diluted mAb against CPS (starting at 10 µg/ml) plus a fixed serum concentration for 30 min at 37°C. Subsequently, bacteria were washed with buffer and incubated with specific monoclonal antibodies for human C1q, C4b, or C3b (all at 1 µg/ml; Quidel) for 30 min at 4°C. Complement components binding was detected with Alexa-conjugated F(ab')2-goat anti-mouse IgG (H+L) (2 µg/ml; Jackson ImmunoResearch Europe Ltd) after 30 min incubation at 4°C. Samples were washed, fixed with 1% ice-cold paraformaldehyde (PFA), and measured in a flow cytometer (BD FACSVerse). Data were analyzed by FlowJo software, and results are presented as relative mean fluorescence index (MFI) compared to the highest concentration of the CPS-IgG-WT.

## Opsonophagocytosis and killing of *S. pneumoniae* by neutrophils

Human neutrophils were freshly isolated from healthy donor blood using the Ficoll-Histopaque gradient method already described (*Bestebroer et al., 2007*; *Boero et al., 2021*). Phagocytosis assay was performed in a round-bottom 96-well plate, and neutrophil-associated fluorescent bacteria were analyzed by flow cytometry. FITC-labeled *S. pneumoniae* strains were opsonized by pre-incubation with twofold serial dilutions of mAb in 5% NHS, or 2.5% IgG/IgM-depleted serum as complement source, in RPMI-HSA or as control in 5% heat-inactivated NHS (30 min at 56°C) for 20 min at 37°C. Subsequently, neutrophils were added in a 1:10 cell to bacteria ratio and phagocytosis was allowed for 30 min at 37°C on a shaker (650 rpm). Ice-cold 1% PFA in RPMI-HSA was added to stop the reaction. Samples were measured by flow cytometry, and mean fluorescence values determined for gated neutrophils (*Boero et al., 2021*). Results are presented as relative mean fluorescence index (MFI) compared to the highest concentration of CPS-IgG-WT. When neutrophil opsonophagocytic killing capacity was assessed, similar procedure was performed with some modifications based on previously described method (*de Jong et al., 2017*). Bacteria were opsonized during 20 min in a round-bottom 96-well plate as described above. For each condition, the mixture was subsequently transferred to sterile none-siliconized 2 ml tubes (Sigma) with neutrophils ($1 \times 10^7$ cells) in a 1:1 ratio in 100 µl volume and the phagocytosis process was prolonged to 45 min at 37°C on a shaker (650 rpm) to ensure intracellular bacterial killing. To release the internalized bacteria, the neutrophils were lysed for 5 min with ice-cold 0.3% (wt/vol) saponin (Sigma-Aldrich) in sterile water. Samples were then serially diluted in PBS and plated onto blood agar plates in duplicate. CFUs were counted after overnight incubation at 37°C 5% (vol/vol) $CO_2$ incubator, and percentage survival was calculated and compared with inoculum.

## Fc-III peptide

The inhibitory Fc-binding peptide (DCAWHLGELVWCT) (*DeLano et al., 2000*) and a scrambled version of Fc-binding peptide sequence, Scr (WCDLEGVTWHACL), were both synthesized by Pepscan (Lelystad, the Netherlands). A fixed concentration (10 µg/ml) was added together with bacteria and sera before the addition of human neutrophils.

## Native mass spectrometry

Native MS experiments were performed a standard Exactive Plus EMR Orbitrap instrument (Thermo Fisher Scientific). Before analysis of protein samples, buffers were exchanged to 150 mM ammonium acetate (pH 7.5) through six dilution and concentration steps at 4°C using Amicon Ultra centrifugal filters with 10 kDa molecular weight cutoff (Merck). For experiments studying the IgG-binding properties of Fc-III and Scr, 1 µM of anti-CD52 IgG1 was incubated with 4 µM Fc-III or 10 µM Scr for at least 15 min at RT. Anti-CD52 IgG1-RGY hexamers were measured at a total IgG1 concentration of 2 µM in the presence or absence of 40 µM Fc-III or Scr. The incubation step with the peptides was proceeded for at least 90 min at 37°C due to the relatively slow disassembly rate of solution-formed IgG1-RGY hexamers. Samples were loaded into gold-coated borosilicate capillaries (prepared in-house) for direct infusion from a static nano-ESI source. Deconvoluted mass spectra were generated by Bayesian deconvolution using UniDec (*Marty et al., 2015*).

## Microscopy

Light microscopy image of pneumococcal internalization by human neutrophils was performed after incubation of *S. pneumoniae* 6B in the presence of 5% IgG/IgM-depleted serum supplemented with CPS6-IgG1 E430G (8 µg/ml), as previously described. Samples were prepared by cytospin (Shandon) and stained with Diff-Quick. Pictures were taken with a Sony Nex-5 camera mounted without lens on an Olympus BX50 microscope using a ×100/1.25 oil objective to visualize cytoplasmic internalization.

For confocal microscopy, samples were then prepared in a glass-bottom cellVIEW slide (Greiner Bio-One [543079]) and incubated with WGA-Alexa 647 for 10 min. Cell VIEW slides were placed in a humid chamber during incubation to prevent evaporation of growth medium. Z-stacks at three random locations per sample were collected at 0.42 µm intervals using a Leica SP5 confocal microscope with a HCX PL APO CS ×100/1.40–0.60 OIL objective (Leica Microsystems). FITC fluorescent bacteria were detected by excitation at 488 nm, and emission was collected between 495 nm and 570 nm. Alexa Fluor 647 fluorescence was detected by excitation at 633 nm, and emission was collected between

645 and 720 nm. Image acquisition and processing was performed using Leica LAS AF imaging software (Leica Microsystems).

## Mice

BALB/c mice from Envigo (Horst, the Netherlands), 8–12 weeks old, that matched for age and sex in each experiment were used. The animals were maintained on a 12 hr light/dark cycle in a room maintained at a mean temperature of 21 ± 2°C with a relative humidity of 50 ± 20%. Drinking water and food pellets were provided ad libitum. Animal experiments were performed at the infection Unit of the Central Animal Facility at Utrecht University and handled according to the institutional and national guidelines for the use and care of laboratory animals. The study was approved by the institutional Animal Care and Use Committee (AVD1150020172204).

## Pneumococcal infection in mice

The pneumococcal infection model has been described before (*Saeland et al., 2003*). In brief, mice were passively immunized intraperitoneally (i.p.) with 200 μl of antibody in PBS, 3 hr before challenge. Mice were anesthetized with inhaled isoflurane at 3% and challenged by intranasal (i.n.) route with $1 \times 10^8$ CFU pneumococci serotype 6A in 50 μl PBS. Every 24 hr after challenge, animals were weighted, scored by two independent researchers, and blood was taken from tail vein. When mice exhibited severe signs of disease, they were sacrificed according to the national guidelines. Every day after challenge, blood was taken from the tail vein. Blood was serially diluted and plated on Colombia agar containing 5% sheep blood (Sanofi Diagnostics Pasteur, Marnes-la-coquette, France). Plates were incubated overnight at 37°C, and pneumococcal colonies were counted. Animals that survived for 7 days were considered protected.

In order to study the effect of administered mAbs locally, the same infection and immunization procedure was followed but mice were sacrificed 24 hr post-infection by an intraperitoneal injection of pentobarbital. Blood, bronchoalveolar lavage fluid (BALF), and lungs were collected to determine bacterial counts. Lungs were weighted for determination of bacterial dissemination, and homogenized in 0.5 ml PBS in tubes containing ±0.2 mm beads (BioSpec) on a tissue homogenizer (MinibeadBeater 24). The number of bacteria was determined by serial dilution as previously described. Bacterial burden in the lungs was determined from the number of CFU present per milligram of tissue.

G*Power version 3.0.10 and PowerSurvEpi_0.1.3 package were used to estimate groups size, aiming for a power of 0.95. A minimum of 12 or 15 mice per group (depending on the assay) was calculated based on the expected difference between CPS6-IgG1-E345K and CPS6-IgG1-WT and experimental variation obtained in a pilot studies. Animal immunization and samples-analyzing investigator was blinded for the injection groups used.

## Enzyme-linked immunosorbent assays (ELISA) for IgG levels

Human antibody concentrations in mouse sera after passive immunization were measured by ELISA using 96-well half-area plates (Corning, #3690) coated with 2 μg/ml sheep anti-human-IgG (ICN, Affipure) in 0.1 M carbonate buffer pH 9.6 for at least 18 hr and blocked with 4% bovine serum albumin (BSA; Serva) in PBS + 0.05% Tween-20. Captured IgG was detected with peroxidase-conjugated F(ab')2-Goat-anti-human-IgG-Fc antibody (Jackson) for 1 hr at 37°C. Reaction was developed using a freshly prepared TMB substrate solution and stopped with 1 M sulfuric acid before determining the $OD_{450}$ using a microtiter plate reader (iMark mic, Bio-Rad).

## Statistical analysis

Statistical analysis was performed with GraphPad Prism software (version 8.3). All data are presented as means ± SD from at least two independent experiments as indicated in the figure legends. Differences in the efficacy of the hexamerization-enhanced variants and the WT antibody across the different concentrations of the dose–response curves were assessed by two-way ANOVA using Dunnett's multiple comparisons test, with individual variances computed for each comparison. For mice survival protection capacity between groups, log-rank (Mantel–Cox) test was used. When two groups were compared, unpaired two-tailed *t*-test was performed.

## Acknowledgements

This work was supported by the Netherlands Organization for Scientific Research (NWO) through the European Union's Horizon 2020 research programs H2020-MSCA-IF (#798032, to LA), the TTW-NACTAR Grant #16442 (to AJRH and SHMR), and European Research Council (ERC) under the European Union's Horizon 2020 research and innovation program (grant agreement no. 101001937, ERC-ACCENT to SHMR). The authors thank Gestur Vidarsson (Sanquin, Amsterdam) and Paul Parren for scientific input and Alexandra Terry (Genmab) for the generated illustrations.

## Additional information

### Competing interests

Frank J Beurskens: FJB, JS, KPMP and SHMR are co-inventor on a patent describing antibody therapies against S. aureus (WO2017198731A1). FJB and JS are Genmab employees. Janine Schuurman: FJB, JS, KPMP and SHMR are co-inventor on a patent describing antibody therapies against S. aureus. FJB and JS are Genmab employees. Kok van Kessel, Suzan HM Rooijakkers: FJB, JS, KPMP and SHMR are co-inventor on a patent describing antibody therapies against S. aureus. The other authors declare that no competing interests exist.

### Funding

| Funder | Grant reference number | Author |
|---|---|---|
| Marie Sklodowska-Curie Actions | 798032 | Leire Aguinagalde Salazar |
| Toegepaste en technische wetenschappen-NACTAR | 16442 | Albert JR Heck |
| Toegepaste en techniche wetenschappen-NACTAR | 16442 | Suzan HM Rooijakkers |
| Horizon 2020 - Research and Innovation Framework Programme | 101001937 | Suzan HM Rooijakkers |

The funders had no role in study design, data collection and interpretation, or the decision to submit the work for publication.

### Author contributions

Leire Aguinagalde Salazar, Conceptualization, Data curation, Formal analysis, Supervision, Funding acquisition, Validation, Investigation, Visualization, Methodology, Writing – original draft, Writing – review and editing; Maurits A den Boer, Suzanne M Castenmiller, Data curation, Formal analysis, Validation, Investigation, Writing – review and editing; Seline A Zwarthoff, Data curation, Formal analysis, Validation, Writing – review and editing; Carla de Haas, Data curation, Validation, Methodology, Writing – review and editing; Piet C Aerts, Data curation, Validation, Methodology; Frank J Beurskens, Conceptualization, Resources, Visualization, Methodology, Writing – review and editing; Janine Schuurman, Conceptualization, Resources, Investigation, Visualization, Writing – review and editing; Albert JR Heck, Data curation, Formal analysis, Investigation, Writing – review and editing; Kok van Kessel, Conceptualization, Data curation, Formal analysis, Supervision, Validation, Investigation, Visualization, Methodology, Writing – original draft, Writing – review and editing; Suzan HM Rooijakkers, Conceptualization, Resources, Supervision, Funding acquisition, Validation, Visualization, Writing – original draft, Project administration, Writing – review and editing

### Author ORCIDs

Leire Aguinagalde Salazar http://orcid.org/0000-0002-4596-8336
Seline A Zwarthoff http://orcid.org/0000-0002-4395-2007
Janine Schuurman http://orcid.org/0000-0002-9738-9926
Albert JR Heck http://orcid.org/0000-0002-2405-4404
Suzan HM Rooijakkers http://orcid.org/0000-0003-4102-0377

### Ethics
Animal experiments were performed at the infection unit of the Central Animal Facility at Utrecht University and handled in strict accordance to the institutional and national guidelines for the use and care of laboratory animals. The study was approved by the institutional Animal Care and Use Committee (AVD1150020172204) and every effort was made to minimize animal suffering.

### Decision letter and Author response
Decision letter https://doi.org/10.7554/eLife.80669.sa1
Author response https://doi.org/10.7554/eLife.80669.sa2

## Additional files

### Supplementary files
• Supplementary file 1. Protein sequences used for antibody production. Variable and constant heavy and light chain protein sequences used for antibody production. The residues E345 and E430 are highlighted in light gray and dark gray, respectively. The adapted amino acids for this study are highlighted in red.

• MDAR checklist

### Data availability
All data generated or analysed during this study are included in the manuscript and supporting file; Supporting data has been uploaded to Dryad.

The following dataset was generated:

| Author(s) | Year | Dataset title | Dataset URL | Database and Identifier |
| --- | --- | --- | --- | --- |
| Aguinagalde Salazar L, Rooijakkers S | 2023 | All_Dataset | https://dx.doi.org/10.5061/dryad.s1rn8pkbt | Dryad Digital Repository, 10.5061/dryad.s1rn8pkbt |

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
