## [Editor Report]

This paper will be of interest to immunologists and infectious disease experts, as it reports the investigation of a novel treatment of invasive pneumococcal diseases using complement-activating monoclonal antibodies. Using a combination of in vitro and in vivo methods, the authors demonstrate convincingly that the introduction of specific mutations in human monoclonal antibodies that target the surface of pneumococcus bacteria can result in enhanced complement activation after these antibodies bind to the bacterial surface.

---

## [Decision Letter]

**Decision letter after peer review:**

Thank you for submitting your article "Promoting Fc-Fc interactions between anti-capsular antibodies provides strong immune protection against *Streptococcus pneumoniae*" for consideration by *eLife*. Your article has been reviewed by 2 peer reviewers, and the evaluation has been overseen by a Reviewing Editor and Carla Rothlin as the Senior Editor. The following individual involved in the review of your submission has agreed to reveal their identity: Sylvia Knapp (Reviewer #1).

Essential revisions:

1) The killing assays should be analyzed according to the bacterial numbers ingested by neutrophils before killing starts. This could be done by performing an assay where the first phagocytosis is allowed (e.g. 20-45min), to then assess the CFUs of a sample of lysed neutrophils at baseline and to then check after short periods like every 5min to assess the ability to kill.

2) in vivo studies should be done showing the effect of administered IgGs on local, lung bacterial loads. This can be either done using the serotype 6B model (if it works), or a model of more progressive lung infection such as the one induced by serotype 3 (for instance ATCC 6303), which is well established and can be titrated to analyze mice after 48h (before any mouse succumbs to the infection).

3) A major concern is that the WT IgG1 / 3 isotypes do not bind complement. Why is this? Is this a Fab effect (ie because of how the antigen is distributed on the organism or affinity) or reflects how the moab are produced, or some other effect? One way to address this is to examine the glycosylation profiles and the complement-fixing activity of anti-capsule isotypes from normal human sera (even if IgM-depleted).

4) Related to this, since the WT antibodies have minimal effects on bacterial killing in vitro (except for 19A at the highest concentration) yet after transfer into mice there is some effect on bacterial levels in the blood (a roughly 2-fold difference in the proportion of mice with detectable bacteria in blood between 100ug WT and E), it is important to examine whether this effect is through human antibodies activating mouse complement as one interpretation is that hexamerisation may not be the sole driver in vivo (eg Figures6c and s9a). Does the adoptive transfer of antibodies into C3-deficient mice have a similar effect? Have the authors examined mouse complement activation by human antibodies? Are multiple mechanisms at play? If Ab to capsule induced after vaccination or infection can be protective yet have a minimal capacity to activate complement then how are they functioning in the absence of enhanced hexamerisation?

---

## [Author Response]

Essential revisions:1) The killing assays should be analyzed according to the bacterial numbers ingested by neutrophils before killing starts. This could be done by performing an assay where the first phagocytosis is allowed (e.g. 20-45min), to then assess the CFUs of a sample of lysed neutrophils at baseline and to then check after short periods like every 5min to assess the ability to kill.

The referees’ comment made us realize that we did not clearly explain our phagocytic killing assays and the role of complement herein. While the complement system is known to enhance phagocytosis (by labeling the bacterial cells with opsonic molecules (opsonization)), it does not play a role in intracellular killing. Our intention with the killing assay was not to claim differences between antibodies after the uptake of bacteria by neutrophils. Our opsonophagocytic killing assay measures all steps leading to killing: opsonization, phagocytosis and intracellular killing. When antibodies induce better opsonization (like the hexamer variants used in this study), this will automatically enhance phagocytosis and the subsequent killing.

For the reader’s understanding, we have corrected this in the manuscript by rephrasing the assay to ‘opsonophagocytic killing assay’ (instead of ‘killing assay’) and added an extended explanation about the assay in the results section (see lines 242-247).

2) In vivo studies should be done showing the effect of administered IgGs on local, lung bacterial loads. This can be either done using the serotype 6B model (if it works), or a model of more progressive lung infection such as the one induced by serotype 3 (for instance ATCC 6303), which is well established and can be titrated to analyze mice after 48h (before any mouse succumbs to the infection).

As suggested by the reviewer, we now include additional new in vivo data to show the effect of administered IgGs on lung bacterial loads. We repeated the mouse experiment following the same immunization and infection procedure. However, now mice were euthanized 24h post-infection to collect bronchoalveolar lavage fluid (BALF) and lung tissue to determine local bacterial loads. No differences were found between local bacterial loads in mice treated with the different monoclonal antibodies. This indicates that mAbs do not prevent bacteria from establishing an infection in the lungs, but mainly affect bacterial dissemination to the bloodstream. These valuable new data are now included in the revised manuscript as *Figure 6—figure supplement 4* and Results section lines 297-304, 565-572.

Of note to the reviewer, all in vivo experiments (including those in the original manuscript) were carried out with serotype 6A, which is known for being more invasive in mice than serotype 6B (see Results section lines 276-279 and reference 66).

3) A major concern is that the WT IgG1 / 3 isotypes do not bind complement. Why is this? Is this a Fab effect (ie because of how the antigen is distributed on the organism or affinity) or reflects how the moab are produced, or some other effect? One way to address this is to examine the glycosylation profiles and the complement-fixing activity of anti-capsule isotypes from normal human sera (even if IgM-depleted).

We agree that this is an important surprise of our study because it was always believed that the binding of an antibody to a target surface alone would be sufficient to trigger Fc-mediated effector functions (like complement activation). Using a well-defined panel of monoclonal antibodies, we here show that wild-type antibodies against encapsulated pneumococci do not automatically trigger complement activation (despite good binding).

We are confident that this is NOT related to production issues. Previously we have generated and functionally characterized monoclonal IgG1/3 antibodies against *S. aureus* (Zwarthoff *et al.,* PNAS. 2021 Jun 29; 118(26): e210278711). The cloning, expression and purification procedures of these anti-*S. aureus* antibodies were identical to the antibodies used in this paper (they even have the same IgG-Fc backbone). Surprisingly the IgG1/IgG3 WT antibodies against *S. aureus* wall teichoic acid (WTA) are well capable of inducing complement activation and downstream phagocytosis (in the absence of Fc-enhancing mutations). Therefore, we believe that the lack of complement activity is not due to how the antibodies were produced. Instead, our data suggest that these anti-capsule antibodies have a poor ability to establish Fc-Fc contacts needed for hexamer formation. This hypothesis is strengthened by our findings that strengthening Fc-mediated hexamerization can strongly enhance weak antibodies. As suggested by the referee, this could be due to the nature of the antigen (density, surface distribution) and/or Fab affinity.

In the revised manuscript, we elaborate on these points more extensively and clarify the differences with our previous data on *S. aureus* (See Discussion line 338 to 351).

Finally, while our paper shows that multiple monoclonal antibodies against capsule have poor complement-inducing capacity, we do not claim that this holds true for all anti-capsular antibodies against pneumococci. Analyses of complex, polyclonal antibody mixtures are out of the scope of our study, which focuses on monoclonal antibodies. Also see Discussion lines 353-355.

4) Related to this, since the WT antibodies have minimal effects on bacterial killing in vitro (except for 19A at the highest concentration) yet after transfer into mice there is some effect on bacterial levels in the blood (a roughly 2-fold difference in the proportion of mice with detectable bacteria in blood between 100ug WT and E), it is important to examine whether this effect is through human antibodies activating mouse complement as one interpretation is that hexamerisation may not be the sole driver in vivo (eg Figures6c and s9a). Does the adoptive transfer of antibodies into C3-deficient mice have a similar effect? Have the authors examined mouse complement activation by human antibodies? Are multiple mechanisms at play? If Ab to capsule induced after vaccination or infection can be protective yet have a minimal capacity to activate complement then how are they functioning in the absence of enhanced hexamerisation?

Thank you for this suggestion. We have performed additional experiments to show that this effect is indeed through antibodies activating mouse complement. Specifically, we now performed phagocytosis assays using mouse serum (BALB/c or CD1 mice) as complement source. First, we observed that human antibodies and mouse complement are a productive combination since human mAbs induced phagocytosis of pneumococci in the presence of mouse complement. Interestingly, we also observed that WT antibodies induce some complement-mediated phagocytosis in mouse serum. This contrasts with the experiments with human sera (figure 2a)*.* We speculate that there are factors in mouse sera that enable some protection by the WT antibody. Most importantly, we found that complement is the driving force for phagocytosis since heat-inactivation (HI) of mouse serum completely abolished antibody-dependent phagocytosis (by WT and hexabody antibodies).

These new data are now included in the revised manuscript as Figure 6—figure supplement 2 and Results section lines 289-295.